Fungi in soil and understory have coupled distribution patterns

Boraks André aboraks@hawaii.edu
Amend Anthony S.
School of Life Science, University of Hawaii at Manoa , Honolulu , Hawai‘i , United States of America
U’Ren Jana
Electronic publication date: 2021 Sep 21
Publication date: 2021
Volume: 9
Electronic Location ID: e11915
Received 2021 Apr 13; Accepted 2021 Jul 14
Copyright: ©2021 Boraks and Amend
Copyright year: 2021
Copyright holder: Boraks and Amend
License: This is an open access article distributed under the terms of the Creative Commons Attribution License, which permits unrestricted use, distribution, reproduction and adaptation in any medium and for any purpose provided that it is properly attributed. For attribution, the original author(s), title, publication source (PeerJ) and either DOI or URL of the article must be cited.
License URL: https://creativecommons.org/licenses/by/4.0/

Keywords: Fungi, Vanuatu, Habitat, Range size, Specialist, Generalist, Soil, Distribution, Spatial, Phylloplane

Funding: National Science Foundation NSF #1555793 André Boraks and Anthony S. Amend were supported by a grant from the National Science Foundation (NSF #1555793). There was no additional external funding received for this study. The funders had no role in study design, data collection and analysis, decision to publish, or preparation of the manuscript.

==============================
Ecological processes that control fungal distribution are not well understood because many fungi can persist in a wide variety of dissimilar habitats which are seldom sampled simultaneously. Geographic range size is reflective of species’ resource usage, and for plants and animals, there is a robust positive correlation between niche-breadth and range-size. It remains unknown whether this pattern is true for fungi. To investigate the fungal niche breadth–range size relationship we identified habitat specialists and generalists from two habitats (plant leaves and soil) and asked whether habitat specialization influenced fungal biogeography. We sampled fungi from the soil and phylloplane of tropical forests in Vanuatu and used DNA metabarcoding of the fungal ITS1 region to examine rarity, range size, and habitat connectivity. Fungal communities from the soil and phylloplane are spatially autocorrelated and the spatial distribution of individual fungal OTU are coupled between habitats. Habitat breadth (generalist fungi) did not result in larger range sizes but did correlate positively with occurrence frequency. Fungi that were frequently found were also found in high abundance, a common observation in similar studies of plants and animals. Fungal abundance-occupancy relationships differed by habitat and habitat-specificity. Soil specialists were found to be locally abundant but restricted geographically. In contrast, phylloplane generalists were found to be abundant over a large range in multiple habitats. These results are discussed in the context of differences between habitat characteristics, stability and spatial distribution. Identifying factors that drive spatial variation is key to understanding the mechanisms that maintain biodiversity in forests.

Introduction

The physical and temporal characteristics of a habitat influence the geographic distribution of its inhabitants (Brown, Stevens & Kaufman, 1996; Clark et al., 2021). Forest ecosystems contain numerous habitats, many of which are found in close proximity and nearly all of which host rich fungal communities. Fungi are cosmopolitan and are found in the soil, rhizosphere, roots, plant litter, foliage, bark, wood, rocks, and water (Větrovský et al., 2020). The particular set of features unique to each habitat drives differences in fungal diversity among habitats (Baldrian, 2017) such that the composition of fungal communities vary predictably even though habitats typically contain a mixture of both habitat specialists and habitat generalists (Amend et al., 2019). Habitat breadth and range size are positively correlated for plants and animals (Slatyer, Hirst & Sexton, 2013) but the correlation between fungal habitat breadth and rarity remains unknown.

Range size is the basic unit of biogeography. Differences in range size among species provide insight into processes of adaptation and dispersal (Pulliam, 2000). Range size and niche breadth represent two of the three major axes of commonness and rarity (Rabinowitz, 1981) and are strong predictors of extinction risk (Chichorro, Juslén & Cardoso, 2019). Brown (1984) argued that species utilizing greater number of resources and persisting among a greater variety of environmental conditions would become more widespread. Brown’s resource-use hypothesis remains a central topic in ecology (Slatyer, Hirst & Sexton, 2013; Sheth, Morueta-Holme & Angert, 2020) but it is seldom examined for microbes, especially fungi. A recent study of plant associated fungi showed that habitat breadth was a strong predictor of the range size over a 5 km gradient (Bernard et al., 2021). These preliminary results suggest that fungi adhere to similar niche breadth –range size correlations that are observed in plants and animals.

As a means for understanding mechanisms of biogeographic patterns, range size lacks the resolution of detailed distribution maps. No standard methodology exists for measuring the area over which a species is found and determining the area of occupancy implies an understanding of the species’ habitat breadth (Gaston & Fuller, 2009). For habitat specialists this may be somewhat straightforward because field sampling can focus on mapping species distributions within a targeted habitat. Fungi are problematic in this regard because species can be recovered from multiple distinct habitats. Within a forest, Aspergillus niger has been isolated from air, deadwood, litter, rhizosphere, root, soil, sediment, and plant shoots (Větrovský et al., 2020). Over 60% of the fungi isolated as endophytes can also be isolated from leaf litter (Osono, 2006), implying that a researcher would need to consider the linked distributions among all potential habitats to effectively measure a species’ range. Linking species distribution maps from more than one habitat provides a novel understanding of how fungi are distributed within and between habitats.

The phylloplane and soil differ in their spatial and temporal dynamics. Leaves have a comparatively short lifespan and can be visualized as habitat islands. Soil, in contrast, is contiguous and can persist indefinitely unless disturbed. Living among both of these habitats are habitat generalists (fungal taxa that occur in multiple habitats) and a set of habitat specialists (fungi that occur exclusively in a single habitat). Soil is one of the most frequently studied forest habitats due to the ubiquity of belowground symbiotic relationships, like mycorrhizae, and the importance of belowground ecosystem processes including carbon and nutrient cycling (Baldrian, 2017). Aboveground, the phyllosphere occupies greater than 1 billion km2; an area more than double the earth’s land surface (Vorholt, 2012). Microbes of the phylloplane modify plant performance and regulate plant diversity, factors that ultimately result in cascading effects on ecosystem functioning (reviewed in (Vorholt, 2012; Griffin & Carson, 2018)). The two habitats have unique and distinct characteristics. Plant leaves have a waxy cuticle layer that reduces moisture and nutrient loss resulting in an oligotrophic surface. Plant leaves experience large fluctuations in diurnal temperature, moisture, wind abrasion, and UV radiation (Vorholt, 2012). By comparison, diurnal moisture and temperature regimes are more stable in the soil. Soil and the phylloplane are separate but adjacent habitats that each harbor rich fungal communities.

Fungi are generally sessile organisms that move about the environment by spore dispersal and hyphal growth. Large and contiguous habitats allow fungal mycelium to propagate for long timespans and great distances. Such is the case for soil dwelling Armalaria ostoyae which have grown to become some of the oldest and largest organisms on earth (Ferguson et al., 2003). It is well established that soil is neither uniform nor static, although given sufficient conditions, soil-dwelling fungi can spread impressive distances via mycelial growth. Despite the potential to spread indefinitely via mycelium, fungi primarily disperse about the environment by spores (both meiotic and mitotic). Unlike mycelial growth, spore dispersal allows fungi to disperse and establish on disconnected habitat patches (Halbwachs & Bässler, 2015), like separate plant leaves. Mycelial growth on plant leaves is restricted in distance by the physical dimensions of the leaf, so it would be advantageous for foliar fungi to also be prolific spore dispersers. Life in the phylloplane offers fungi exceptional access to the vectors that disperse spores, but because of the discontinuity and ephemerality of the habitat, dispersal is uncertain. In comparison to the phylloplane, hypogeous soil-dwelling fungi have access to a habitat that is larger more and temporally stable and as a result have the potential to spread great distances via mycelial growth or even remain dormant in the spore bank for many years (Bruns et al., 2009; Nguyen, Hynson & Bruns, 2012). Previous research comparing soil and airborne fungal communities found soil communities were primarily correlated with abiotic parameters, compared to airborne communities that shifted temporally (Kivlin et al., 2014). Differences in parameters affecting communities in these two habitats brought us to question whether habitat type influenced fungal range size.

We suspected that fungal communities from the phylloplane and soil to be spatially related because phylloplane fungi are transported vertically by senescent leaves that fall to the soil below. In this study we present a spatial perspective on the similarities among the phylloplane and soil mycobiomes. If there is community membership overlap among the soil and phylloplane, is the overlap spatially autocorrelated? Range size has is often correlated with niche breadth (Slatyer, Hirst & Sexton, 2013; Bernard et al., 2021) so we investigated the relationship between habitat specificity and range size. Range size can refer to a number of different measurements (Brown, Stevens & Kaufman, 1996), so for the purpose of this study, range size is the geographic area occupied by an OTU within a transect. Do the range sizes of habitat generalists and habitat specialists differ for leaves and soil? Finally, does habitat type influence range size? Forest dwelling fungi occupy a variety of different habitats, here we explore the relationship between habitat occupancy, connectivity, and fungal biogeography.

Materials & Methods

Study site

Field sampling consisted of two separate forays in August 2017 (Aneityum) and in December 2017 (Tanna) in the province of Tafea, Vanuatu as previously described (Boraks et al., 2021). The two islands are separated by 86 km of open ocean and share a similar tropical climate. Aneityum receives 2,322 mm of rain annually and has an average annual temperature of 19.7 °C, although there is large seasonal and interannual variation (Vanuatu Meteorology and Geo-hazard Department, Australian Bureau of Meteorology, CSIRO, 2015). Our sampling locations are classified as low to mid-elevation rain forest (Mueller-Dombois & Fosberg, 2013) dominated by Syzygium, Dysoxylum, and Hedycarya species (Boraks et al., 2021).

Six forest transects (10 m by 40 m) were sampled, three on each island (Fig. S1). Within each transect we sampled 36 locations in grid formation. At each sampling location, we collected the fungal community from both the soil and the leaf surface (phylloplane). Fungal communities were harvested using flocked sterile swabs and preserved in CTAB buffer. Soil samples were collected by brushing away loose plant litter and inserting swabs into the organic layer of topsoil. Phylloplane samples were collected by swabbing leaf surfaces. Leaf sampling height varied by availability, but generally were located between 1 and 3 m above the forest floor. The total area sampled for each phylloplane sample was constrained to a surface area of 400 cm2, or roughly the surface area of two hands. Plant taxonomy influences the phyllopsphere mycobiome to a greater (Kembel & Mueller, 2014), or lesser (O’Rorke et al., 2015), degree and we attempted to reduce this variance by swabbing leaves from multiple plant species using the same swab. We aimed to include at least three different plant species per sampling site.

Soil and phylloplane samples were taken in pairs from the same geographic location, such that the phylloplane-sampling grid and the soil-sampling grid were coupled. This paired-habitat and spatially explicit sampling scheme totaled 432 fungal community samples (36 sampling sites × 2 habitats × 6 transects) (Boraks et al., 2021).

DNA extraction, PCR and sequencing

Sequencing library preparation was previously described in Boraks et al. (2021). DNA was extracted from the swabs using the Qiagen DNeasy PowerSoil DNA Isolation Kit (Qiagen, Maryland). We slightly modified the extraction protocol by extracting directly from swabs. Amplicon libraries were prepared in a single PCR reaction using Illumina-barcoded fungal-specific primers as in Amend et al. (2019). The primers ITS1F and ITS2 were used to target the hypervariable nuclear ribosomal ITS1 region which is flanked by the 18S and 5.8S rDNA regions (Schoch et al., 2012). Sequence library was purified and normalized (Just-a-plate; Charm Biotech), quantified by qPCR and sequenced on Illumina MiSeq platform using V3 chemistry (Illumina Inc., San Diego, CA) at the Institute for Integrative Genome Biology (University of California–Riverside). Additional details of sequence library prep can be found in Boraks et al. (2021). Sequences are deposited in the NCBI Sequence Read Archive (BioProject: PRJNA634909).

ITS1 sequences were extracted from the flanking ribosomal subunit genes using ITSxpress (Rivers et al., 2018), then filtered by quality scores using the FASTX-Toolkit (Gordon & Hannon, 2010). Reverse sequence reads were discarded because of lower quality reads. Chimeras were detected and removed using vsearch (Rognes et al., 2016). Sequences were clustered at 97% identity and fungal taxonomy was assigned using the Python package constax (Gdanetz et al., 2017) and the Unite 8.0 database. Putative contaminants were identified based on their prevalence in extraction and PCR negative controls and removed using the R package decontam (Davis et al., 2017). OTUs that could not be assigned to a fungal phylum were discarded. The sequence library was normalized by variance stabilizing transformation (VST) (McMurdie & Holmes, 2014) in DESeq2 (Love, Huber & Anders, 2014) within R (R Core Team, 2019). Singletons were removed for all analyses except those that measured community dissimilarity. Sequence read abundance data, taxonomic assignments, sample metadata, and ancillary collection data were compiled with the R package phyloseq (McMurdie & Holmes, 2013). Using sequence read abundance as a metric for species abundance is contentious due to the compositional nature of high-throughput sequencing datasets (Gloor et al., 2017). Statistical analyses for this study rely on both presence-absence and sequence abundance data. Caution should be exercised when interpreting analyses that use relative sequence read abundance as metric for OTU abundance.

Statistical analyses

Are fungal communities different between habitats?

Phylloplane and soil fungal communities were compared using NMDS ordination (using Bray–Curtis dissimilarity) and permutational analysis of variance (permanova) as implemented in the Vegan package (Oksanen et al., 2013). The permanova (adonis) was iterated with 999 permutations.

Inter-habitat spatial autocorrelation

We tested for spatial autocorrelation in OTU co-occurrence between habitats. Bray–Curtis community dissimilarity measurements assessed whether samples collected from different habitats, but from the same geographic location, were more similar than pairwise draws of disjunct locations and habitats. Our null hypothesis was that differences in community composition were independent of habitat and geographic distance. This analysis was conducted in a pairwise manner and a student t-test was calculated. The analysis was then performed at two different spatial scales. The two spatial scales tested differed in grain size: the first tested for inter-habitat spatial autocorrelation by comparing point-samples across the extent of our whole study (both islands ∼110 km), whereas the second test measured spatial autocorrelation between point-samples and the extent of a transect (40 m).

Area of occupancy

To compare range sizes between habitats (soil or phylloplane) and lifestyle (specialist or generalist), we first categorized OTUs by habitat and lifestyle. Habitat specialization was assigned if an OTU was found more than once and exclusively in a single habitat type, whereas habitat generalists occurred more than once in each habitat from the same transect (Fig. S2). Occupancy was calculated by the number of samples in which an OTU occurred within a transect. This analysis was constrained within transects so the maximum occupancy was limited to the number of sampling points within a transect (36 locations, or roughly 400 m2). A Wilcoxon rank sum test, corrected for multiple hypothesis testing (Benjamini & Hochberg, 1995), was used to compare median range size.

Range size per substrate and specialization

To measure the range size of individual OTUs, we identified those that were spatially autocorrelated using a Mantel test with Bray–Curtis distances based on OTU relative sequence-read abundance and then built empirical semi-variogram models for each spatially autocorrelated OTU. Semi-variogram models are useful for analyzing the range of variation between geospatial phenomena. In this study we used semi-variograms to measure the geospatial decay of OTU relative sequence read abundance. Semi-variogram models were developed for each spatially autocorrelated OTU and geographic-range sizes were estimated from the range of each variogram model. The nugget (zero) and model type (exponential) were fixed for all variograms to increase comparability of range size between models. Range sizes were aggregated by habitat and specialization and then compared by pairwise Wilcoxon tests corrected for multiple hypothesis testing. Kriged surface maps were paired for the generalist OTU that occurred in both habitats of the same transect.

Occupancy-abundance relationships

To compare OTU rarity, we generated occupancy-abundance (OA) regressions for all OTUs of each habitat and then compared linear regressions (loess fit) for specialists and generalists of the soil and phylloplane. OTUs were first categorized by habitat and specialization, and then the number of occurrences per transect was tallied for each OTU. This type of OA regressions is known as interspecific occupancy-abundance relationships because the regressions compare multiple OTUs. The maximum occupancy of any single OTU is 36, because there were 36 sampling locations per transect. Abundance was measured using normalized sequence reads.

Results

Are fungal communities different between habitats?

When all transects were combined, 4949 OTUs were identified in the soil and 5362 in the phylloplane. Ten OTUs were previously identified as suspect contaminats and were removed from the dataset. Transects contained an average of 4415 OTUs (standard error of mean (SE) ± 340; n = 6). On average, 55% (SE ± 7%) of the OTUs that were present in a transect could be found in the soil habitat. This was similar for phylloplane, in which 65% (SE ± 5%) of the OTUs that were present in a transect could be found in the phylloplane. Permanova indicated significant differences (p <  0.001) in community composition between soil and phylloplane (habitat R2 = 0.11, p < 0.001) and island (R2 = 0.03, p < 0.001; Fig. 1A).

Figure 1 A comparison of soil and phyllosphere fungal communities.

NMDS ordination (A) indicating differences between fungal communities based on Bray–Curtis dissimilarity. Each point is a fungal community sequenced from the soil (circle) and phylloplane (triangle). Ellipses represent 95% confidence intervals around the centroids. Separation along axis 2 corresponded to difference in islands and sampling date. Permanova: habitat (R2 = 0.11, p < 0.001) and island/date (R2 = 0.03, p < 0.001). Venn diagram (B) emphasizing the distribution of OTU generalist and specialists for each habitat (observed richness). Singleton OTU numbers are excluded from diagram.

Inter-habitat spatial autocorrelation

Composition of fungal communities of the soil and the phylloplane differ, yet 22% of OTUs co-occur in both habitats (generalists). Co-occurrence between habitats was spatially autocorrelated at a large scale (110 km), phylloplane communities were more similar to proximate soil communities than distant soil communities (Table 1). At a smaller scale (<40 m) the spatial autocorrelation of fungal communities from different habitats is less supported. Inter-habitat spatial autocorrelation was statistically significant for three of the six transects (Table 1).

Table 1 Inter-habitat dissimilarity measurements comparing paired soil-phyllosphere communities and a null hypothesis.

Spatial autocorrelation is indicated by the smaller dissimilarities between observed communities than the null-hypothesis. Significance was tested by student t-test. The entire study spanned < 110 km, and the six transects (T1–T12) each covered an area of 400 m2.

		Transect	
	Entire study	T1	T4	T6	T9	T10	T12	
Distance	(<110 km)	(10 × 40 m)	
	Bray–Curtis dissimilarity (Soil-Phyllosphere)	
Observeda	0.925 ∗∗	0.926	0.881	0.934 ∗	0.929 ∗	0.928 ∗∗∗	0.960	
Randomizedb	0.958 ∗∗∗	0.931	0.893	0.948 ∗	0.940 ∗	0.943 ∗∗∗	0.964	
Notes.

a mean BC dissimilarity for fungal communities from soil and phyllosphere of the identical sampling site.

b mean BC dissimilarity for fungal communities from soil and phyllosphere of a randomized sampling site.

p-value: <0.001∗∗∗; <0.01∗∗∗; <0.05∗.

Area of occupancy

An OTU was categorized as a habitat specialist if it was found more than once and exclusively among a single habitat type. We identified 2,187 soil specialist OTUs and 2,212 phylloplane specialist OTUs (Fig. 1B). We defined OTU generalists as those occurring more than once in each habitat. A total of 1307 unique OTUs were classified as generalists. Thirty-five percent of the generalist OTUs occurred in more than one transect. Generalist fungi had higher occupancy rates than specialist fungi (Fig. 2). Median occupancy rates were greater for generalist fungi (phylloplane generalist mdn = 6, iqr = 12; soil generalist mdn = 4, iqr = 6) than for specialist fungi (phylloplane specialist mdn = 3, iqr = 3; soil specialist mdn = 3, iqr = 4) (Fig. 2).

Figure 2 Transect occupancy measured by the number of times an OTU occurred within a transect.

Histogram bars represent a tally of the OTU density for particular transect occupancy levels (y-axis). Occupancy (x-axis) is measured as the number of occurrences per transects (max 36). Generalist fungi occur more commonly than specialist fungi as is indicated by the proportion of generalists found on the right-hand side of the histogram. The number of occurrences for each habitat and specialization pair differed significantly (p < 0.001) as determined by pairwise Wilcoxon test corrected for multiple hypothesis testing. Y-axis is on a Log10 scale.

Semi-variogram modeling

Spatial autocorrelation was detected among 7.7% of the phylloplane specialists and 10.4% of the soil specialists. Spatial autocorrelation was detected among 10.5% of the phylloplane generalists and 18.8% of soil generalists. Range sizes were modeled using experimental variograms that were based on the number of sequence reads as a function of spatial distance. A total of 1,060 variogram models were generated. Variogram models were manually inspected, and we culled models (515/1060) that reported unrealistically small ranges (<1 m) for our given sampling scheme. In addition, 6 OTUs were identified as outliers and, despite robust variogram models, had exceptionally large ranges and were removed from downstream analysis (Table S1). The six outliers with large range sizes (mean = 737 m) included five OTUs within the Ascomycota and one OTU within Glomeromycotina. The removal of unreliable models and outliers resulted in 539 variograms remaining. Range size of specialist soil fungi (2.78 m; iqr 1.77) were significantly greater than phylloplane generalist fungi (2.11 m; iqr 1.53) and phylloplane specialist fungi (2.02 m; iqr 1.82) (p < 0.05). Soil generalist fungi (2.54 m; iqr 1.66) did not significantly differ in range size from soil specialists or phylloplane generalists (Fig. 3). Inclusion of the culled outliers OTU did not affect the significance of range-size comparisons. Spatially autocorrelated OTU sequence read abundance generally become insignificant around at distances greater than 5 m, although several ranges were estimated to be hundreds or thousands of meters.

Figure 3 Range size (meters) for individual fungi of the soil and phyllosphere as calculated from the range of semi-variogram models.

Specialist fungi occurred more than once and exclusively within a single habitat, whereas Generalist fungi were found in both habitats. While the identities of soil and phyllosphere generalist OTUs are the same, the range size estimated for each OTU differs per habitat. Each dot represents the range size for a single fungal OTU. The colored areas indicate the probability distribution of range sizes. OTU abundance is measured as a function of relative sequence read abundance. Outliers with large range sizes are not presented here but can be found in Supplemental Table S1.

Distribution maps for generalist OTU

Kriged surface maps were produced for 11 OTUs that were identified as spatially-autocorrelated generalist fungi that occurred in the same transect. The OTUs were identified as OTU97_1160_Dothideomycetes sp.; OTU97_1992_Helotiales sp.; OTU97_2240_Agaricomycetes sp.; OTU97_228_Psathyrella sp.; OTU97_257_Polyporale sp.; OTU97_2608_Xylariales sp.; OTU97_2833_Agaricales sp.; OTU97_3832_Psathyrellaceae sp.; OTU97_3832_Psathyrellaceae sp.; OTU97_5320_Xylariales sp.; OTU97_6890_Ascomycota sp. In general, the spatial distribution of an OTU was synchronous between habitats (Fig. 4). Semi-variogram models and kriging standard errors are reported in the supplemental (Fig. S3)

Figure 4 Generalist fungi that occur in soil are often found nearby in the phyllosphere.

Maps detailing the distribution of 11 fungal OTUs (A–K). Each set of maps illustrates the distribution of OTU sequence read abundance within a transect (10 m by 40 m). Maps are coupled so that the distribution of an OTU from soil is paired with the same OTU from the phyllosphere of the same transect. Dark colors indicate low abundance and light colors indicate high abundance. The OTUs were identified as: (A) Dothideomycetes sp.; (B) Helotiales sp.; (C) Agaricomycetes sp.; (D) Psathyrella sp.; (E) Polyporale sp.; (F) Xylariales sp.; (G) Agaricales sp.; h. Psathyrellaceae sp.; (I) Psathyrellaceae sp.; (J) Xylariales sp.; (K) Ascomycota sp.

Occupancy-abundance relationships

Phylloplane and soil occupancy-abundance relationships followed classical biogeographical expectations; the number of samples in which an OTU occurred, was positively correlated with the mean abundance of that OTU (Fig. 5). Among the phylloplane, specialists and generalists had similar occupancy-abundance regression slopes at low occupancy levels, but the regressions diverged at high occupancy levels (Fig. 5). Common phylloplane generalists were more abundant than common phylloplane specialists. Among the soil, low-occupancy specialist fungi were more abundant than low occupancy generalists, but specialist and generalist abundance converge at higher occupancy rates (Fig. 5).

Figure 5 Interspecific occupancy-abundance relationships for OTUs of the soil and phyllosphere.

Soil specialists are more abundant than soil generalists at low occupancy levels. Phyllosphere specialists and generalists diverge at high occupancy levels. Occupancy (x-axis) is the number of sampling sites, within a single transect, that an OTU was found. Local abundance (y-axis) is the sum of an OTU’s sequence read abundance (VST corrected) divided by the occupancy rate. Regression fitted with a loess curve and shaded ribbons show the 95% confidence interval.

Discussion

Co-occurrence among habitat types

Here, we report a first attempt to couple distribution maps of fungal OTUs in the soil and understory. These map stacks provide a rich visualization of where generalist fungi persist simultaneously in either habitats across the forest landscape (Fig. 4). In addition to the visual evidence, inter-habitat spatial autocorrelation confirmed that fungal communities of the phylloplane are more likely to be similar to nearby soil communities than to distant soil communities (Table 1). The inter-habitat spatial autocorrelation observed here was especially pronounced at large spatial scales (<110 km) but became less evident at smaller spatial scales (<40 m). Stated differently, this result indicates the possibility of predicting the geo-location of a leaf surface based on the fungal community of soil samples (or vice versa), and that the accuracy of this prediction becomes less reliable at small spatial scales (<40 m).

Migration between habitats is important for community turnover of ephemeral habitats such as pools of water, phytotelma, and plant leaves. Leaf age is directly correlated with mycobiome composition. Young, newly emergent plant leaves are sparsely colonized by fungi (Wildman & Parkinson, 1979; Suryanarayanan & Thennarasan, 2004; Oono et al., 2015). The initial colonization is a stochastic processes via horizontally transmitted fungi from unknown environmental sources (Unterseher et al., 2018). As the leaf ages, fungal infection increases and beta-diversity decreases (Oono et al., 2015). The decrease in beta-diversity may be partially driven by host identity (Unterseher et al., 2013; Kembel & Mueller, 2014; O’Rorke et al., 2015). We attempted to compensate for the effect of host identity by sampling the leaves of multiple plant hosts from the same sampling location. After a leaf has senesced and fallen from the plant, it will carry a rich microbial community to the forest floor. The phylloplane and soil are physically disjunct habitats but spatially autocorrelated overlap in their mycobiomes (Table 1) is suggestive of migration between them.

It is well established that microbes from the forest floor migrate to the phylloplane and vice versa. Indeed, this is the main tenant of the latent saprotroph hypothesis which proposes that a saprotroph is able to colonize the living tissue of its host to gain competitive advantage when the host tissue begins to senesce (Porras-Alfaro & Bayman, 2011; Peršoh, 2013). Phytopathology literature contains examples of foliar pathogens that overwinter in ground litter, only to migrate back to the phylloplane when conditions are optimal. For example, the tree pathogen Venturia inaequlis is a hemibiotroph and must spend part of its lifecycle as a parasite, and part of its lifecycle on the dead tissue of the same host as a saprotroph (Agrios, 2005). In line with our results (59%, n = 3,519), a previous report examining the overlap of phyllosphere and leaf litter found 64% of leaf epiphytes could also be isolated from leaf litter (Osono, 2006). An important distinction in our study is that we sampled from the organic soil layer rather than leaf litter. Evidence from bacterial (Bai et al., 2015; Zarraonaindia et al., 2015; Wagner et al., 2016) and fungal systems (Amend et al., 2019; Bernard et al., 2021) have shown that soil may act as a reservoir of microbes that migrate to the phylloplane, an observation that is supported by our results (Table 1).

Habitat breadth and range size

Range size and niche breadth are positively correlated among plants and animals (Slatyer, Hirst & Sexton, 2013; Bernard et al., 2021). Here we show that for fungi, niche breadth may or may not correlate with range size, and that this equivocal result depends on how range size is calculated (Figs. 2 and 3). Range size and habitat specificity are two of the major axes of rarity (Rabinowitz, 1981) and by including sequence read abundance we explore commonness and rarity of fungi from the phylloplane and soil.

There is no standard methodology for measuring species’ geographic range. The large suite of techniques available can capture fundamentally distinct features of species distribution (Gaston, 1996). We used two different approaches to measure fungal geographic range size. First, we considered area of occupancy per transect (Fig. 2). This is a tally of the number of sampling-sites at which an OTU is recorded within a transect. Calculating range size by areas occupied is advantageous because of its simplicity, it requires only presence-absence data and it is inclusive of rare species (Gaston, 2008), and was successfully used in a study of bacterial range size (Choudoir et al., 2018). The drawback to this approach is that it lacks distribution nuance; it is unable to differentiate between a sparse and clustered distribution of the same occupancy level. The second method we used to measure range size was variography which provided a measurement most similar to Gaston’s (1996) geometric circle (Fig. 3). In this study the radius a range was calculated from the range of an empirical semi-variogram. This technique was previously used in a study of ectomycorrhizal fungi (Pickles et al., 2010). Another way to think of this measurement is to consider the linear distance, from the geometric center of a range, at which variation in OTU sequence abundance is no longer spatially autocorrelated. The advantage of the variogram approach is that it provides a linear distance measurement which can be used to make visually illustrative maps (Fig. 4). When interpreting kriged variogram maps (Fig. 4) generated using high throughput sequencing data, it is important to consider their limitations. Variogram modeling is only possible on a small fraction of our metabarcoding dataset (Fig. S2) because of the large amounts of spatially autocorrelated data required and most OTU datasets are dominated by low-abundance taxa. Relative sequence read abundance is compositional and therefore comparisons between samples may be misleading (Gloor et al., 2017; Morton et al., 2019). Additionally, amplicon sequencing is indiscriminate and we are unsure whether we sequenced spores, hyphae, or exogenous DNA (Carini et al., 2016).

Brown (1984) affirmed that species which have a broad environmental range and are able to use a wide range of resources will be both locally abundant and widespread. For fungi, adjacency of steppingstones within a habitat matters little if you can just as easily step between habitats. In an analysis of soil bacteria, Luo et al. (2019) found that generalist bacteria occurred at greater abundances than specialist bacteria. Results from our area of occupancy analysis support theoretical expectations insofar as we found that generalist fungi occur more frequently than specialists (Fig. 2). Interestingly, greater transect occupancy levels did not translate into larger modeled range sizes. Variogram estimations indicated no significant differences in range sizes between habitat specialists and generalists (Fig. 3). Lack of difference in the variogram range sizes might be attributable to the insensitivity of variograms to low occupancy OTU. Differences in range size results from each technique highlights the complicated nature of spatial distribution and habitat specialization.

A positive interspecific abundance–occupancy relationship is one of the most widely observed patterns in macroecology. We tested for variation in abundance-occupancy regression slopes between specialists and generalists from two separate habitats. OTUs of this study exhibited a positive occupancy-abundance relationship which describes the greater-than-proportional increase in abundance with respect an increase in occupancy (Fig. 5). When transect occupancy is low, soil specialists are more abundant than soil generalists (Fig. 5). We believe this may be result of the comparative stability of soil. Without significant perturbation, soil microbial communities show limited turnover (Kivlin et al., 2014; Žifčáková et al., 2016) potentially due to the longevity of spore banks (Bruns et al., 2009; Nguyen, Hynson & Bruns, 2012). In less disturbed habitats, a strong competitor can persist in one location, cumulatively adding to the volume of spores within the spore bank of that location. This process might explain the pattern of locally abundant soil specialists towards the left side of Fig. 5B. The result of this study are similar to a previous report that belowground fungi had high local abundance but small species ranges, while aboveground fungi had lower local abundance but larger spatial ranges (Kivlin et al., 2014). Generalists occur more commonly and abundantly among the phylloplane, whereas in the soil specialist fungi were found at high abundance but low occupancy (Fig. 5). By contrasting the physical and temporal characteristics of these two habitats we were able to infer potential strategies employed by fungi to persist on the landscape.

Conclusions

Geographic range size is a basic unit of biogeography, and its positive correlation with niche-breadth is a well-established phenomenon observed among plants and animals. The importance of this correlation cannot be understated; it is proposed as a mechanism for rarity, has implications for extinction risk, and may provide insight into processes of adaptation and dispersal. We demonstrate inter-habitat spatial autocorrelation of fungal communities, an indication of fungal dispersal between soil and the phylloplane. Additionally, fungi of differing habitats have similar biogeographic patterns. Local range size for fungal OTUs rarely exceed 5 m. Habitat generalists and specialists have different occupancy-abundance models, an indication that the effects of specialization on range size is habitat dependent. The phylloplane and soil are habitats with dissimilar temporal and environmental characteristics and this study provides important insights of fungal distribution within and between these two habitats. Phylloplane generalist fungi have larger range sizes than phylloplane specialist fungi. This not true for soil dwelling fungi. Soil specialization is associated with smaller range sizes and greater abundances. Fungal biogeography is determined by the unique spatial and temporal characteristics of the habitat.

Supplemental Information

Supplemental Information 1 Map of sampling location in Vanuatu with transect locations

Red squares on map indicate transect locations (a). Each transect (10 m by 40 m) had 36 sampling locations, indicated by the red circles on the illustrated transect (b).

Click here for additional data file.

Supplemental Information 2 Flowchart detailing identification of habitat specialist and habitat generalist fungi and the selection process used identify OTUs included in Fig. 4

Click here for additional data file.

Supplemental Information 3 Semi-variogram models, error estimation, and kriged surfaces

Generalist OTUs of the same transect were selected by protocol (Fig. S2) and mapped using kriging. This supplemental contains a compilation of the semi-variogram models, error estimation, and kriged surfaces associated with maps published as Fig. 4.

Click here for additional data file.

Supplemental Information 4 Excel table containing taxonomic classification of OTUs and variogram-estimated range size

This table includes the 6 OTUs that were deemed outliers.

Click here for additional data file.

This project would not be possible without the hard work and dedication from the ever-growing network of people associated with Plants mo Pipol blong Vanuatu. A special thanks for assistanct from Presley Dovo, Frazer Alo, Thomas Doro, Stephanie Sali, Jean-Pascal Wahe, the late Philemon Ala, Marika Tuiwawa, Alivereti Naikatini, Sean Thackurdeen, Jonathan del Rosario, Chanel Sam, Ashley McGuigan, Tamara Ticktin, Tom A. Ranker, Gregory M. Plunkett, and Michael J. Balick. We are extremely grateful to the many communities on Aneityum and Tanna for their kindness, hospitality, and commitment to this project, tankyu tumas. The authors would like to thank the handling editor Jana U’Ren and reviewers Justin Shaffer, Elizabeth Bowman, and anonymous for thoughtful reviews. Nicole Hynson, Tom A. Ranker, Nhu Nguyen, Mikey Kantar, contributed to project design and manuscript improvement. We are also very appreciative of The Vanuatu Department of Forestry for their partnership and logistical support. This study is part of the NSF project, “Collaborative Research: Plant and Fungal Diversity of Tafea Province, Vanuatu, A Threatened Pacific Hotspot. This is publication 106 from the School of Life Sciences, University of Hawai’i at Mañoa.

Additional Information and Declarations

Competing Interests

Author Contributions

DNA Deposition

Data Availability

Anthony Amend is an Academic Editor for PeerJ.

André Boraks conceived and designed the experiments, performed the experiments, analyzed the data, prepared figures and/or tables, authored or reviewed drafts of the paper, and approved the final draft.

Anthony S. Amend conceived and designed the experiments, performed the experiments, analyzed the data, authored or reviewed drafts of the paper, and approved the final draft.

The following information was supplied regarding the deposition of DNA sequences:

The sequencing dataset is available in the NCBI Sequence Read Archive (BioProject: PRJNA634909).

The following information was supplied regarding data availability:

The R code used for analyses in this article is available at GitHub: https://github.com/aboraks/Mapping_Vanuatu_OTU.git.

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
