# Peer review of "Fungi in soil and understory have coupled distribution patterns"

_PeerJ, doi:10.7717/peerj.11915_

## Round 0.1 · original submission · Minor Revisions

All three reviewers have commented very positively on your study and the manuscript. However, they identified a number of issues that require clarification, such as the choice of metric to estimate range size and methods for normalization, which are central to your conclusions.

In addition, the lack of consistent terminology should be addressed. In addition to Reviewer 3's comments in this regard, I would also add that (i) soil and litter are not interchangeable terms and (ii) the use of the term phyllosphere here is misleading, as this encompasses both the surface and interior of leaves. Phylloplane is a more appropriate term for this study as fungi were obtained only from the leaf surface. This also explains why you would observe more overlap with soils compared to previous studies on endophytes. Thank you for fully addressing all of these suggestions. I look forward to reading your revised manuscript.

Reviewer 1 ·

Basic reporting

Please see the review report.

Experimental design

Please see the review report.

Validity of the findings

Please see the review report.

Additional comments

In this study, Boraks and Amend conducted a survey of fungi in soil and phyllosphere in a tropical forest. They found that (a) soil and phyllosphere had different fungal community composition; (b) generalist fungi (present in both soil and phyllosphere) did not have larger range sizes; (c) soil specialists were locally abundant but phyllosphere generalists were abundant over a larger geographic range.

This manuscript is well written. In addition, it contributes to a growing body of literature on microbial dispersal. I also found that the figures are aesthetically pleasent. However, their method of range size calculation needs to be justified and some statements need to be clearly clarified. Below, I provide some comments that outline my major concerns and provide recommendations to hopefully improve the manuscript.


Major comments:

The authors used semi-variogram to calculate range size. However, this method is usually used to calculate spatial autocorrelation range rather than range size. In ecology, area of occupancy (AOO) and extent of occurrence (EOO) are more widely used to calculate range size (Choudoir et al., 2018). Citations are needed to justify the reasons why they used semi-variogram to calculate range size.

Choudoir, M.J., Barberán, A., Menninger, H.L., Dunn, R.R. & Fierer, N. (2018). Variation in range size and dispersal capabilities of microbial taxa. Ecology, 99, 322–334.

L131-132: A major limitation of this study is that plant species identity information was not included in the analyses. This is particularly relevant for phyllosphere-associated fungi. For example, in Figure 1 a certain proportion of variation in community composition of phyllosphere-associated fungi can be explained by plant species identity. Could the authors add this information into their analyses? If not, please address it in the discussion.

L189-19, Figure 2 and Figure 5: Why was the occupancy calculated at the within-transect level? Wouldn’t it ignore broad-scale dispersal patterns, for example, inter-transect and inter-island dispersal? This needs to be justified. I question whether their results and conclusions still hold when looking at larger spatial scale.

Figure 4: Please report the cross-validation results of kriging. That is, the accuracy of spatial kriging prediction.


Minor comments:

L224-226: Please report the R2 from PERMANOVA.

Figure3: Ground and understory? Maybe soil and phyllosphere? Also, I don’t see there are any shaded areas.

In some places “p-value” was used, while in other places “P” was used. Please make it consistent throughout the manuscript.

·

Basic reporting

Introduction
- line 90 - not sure if 'spore' should be 'spores' or 'spore dispersal', or was desired as is.
- line 95 - I think it would be helpful to briefly define Xenospores here (e.g., 'Xenospores (i.e., spores that are dispersed vs. those that stay where they are produced).').

Methods
- line 147 - My understanding is that the MiSeq and NextSeq are unique Illumina instruments/platforms (see https://www.illumina.com/systems/sequencing-platforms.html). Can you please clarify which platform was used? Also, just FYI, I've noticed that for Illumina instruments with two-color chemistry, such as the NextSeq and NovaSeq, introduce high-quality, poly-G tails of various lengths to R1 reads, which are often not detected but can affect downstream analyses. So, depending on which platform was used, this can be relevant.

Results
- Figure 1A - I suggest to add outlines to sample points/triangles, as well as an alpha value to add transparency so that the number of points and their overlap is more clear. I also suggest to include the results from PERMANOVA here if relevant.

Experimental design

Methods:
- line 157 - More and more I see debate over the use of deseq2's VST vs. other sample normalization methods. I'm fine with it's use here (i.e., except where noted below re: Fig. 5), but just want to highlight this and suggest you continue to explore novel methods for normalization as they develop.
- line 165 - Similar to above, more often today I see debate over the use of Bray-Curtis distance outside of analyses of beta-diversity where the goal is to emphasize differences based on 'rare' taxa. Issues surrounding the sparse- and compositional-nature of these type of data point to using pseudo-counts/matrix completion and/or compositionally-aware diversity metrics to avoid this emphasis, which apparently stems in part from BC's inability to handle the sparsity issue. Again, your results are likely robust to this and I'm fine with the use of BC here, but I recommend to explore a new metric called robust Aitchison that can be implemented in the deicode package in qiime2. This metric is compositionally aware and uses matrix completion to deal with the sparsity issue. There is also a phylogenetically-aware version, which is my second comment that you may consider using a phylogenetically-aware distance metric, such as unifrac. I understand the difficulty in constructing fungal phylogenies with short-read ITS data, but there may be a way to use fragment insertion into a larger backbone context, to get a tree that can be used.
- line 165 - Did you explore permdisp in parallel to permanova? It is typically not appropriate to interpret the results of permanova if the results of permdisp are significant i.e., indicate differences in group variances.
- line 195 - The other analyses using VST normalized read counts above appear to be based on presence/absence alone and thus are appropriate. However, as I describe further below for the analyses in Figure 5, any analysis using relative read counts should consider that the read count data are compositional (e.g., you can not tell, from sample-to-sample, if taxon A's abundance is going up vs. taxon B's abundance going down). One solution is to use reference frames (see https://www.nature.com/articles/s41467-019-10656-5). I highly recommend you consider this to accurately estimate these range sizes.
- line 211 - The same concern I have for your range size estimates holds here - please exert caution in using raw/VST-normalized read counts for comparing relative abundances among groups.

Results:
- Figure 5 - I suggest to examine the log-ratio of local abundances of generalists to specialists, for technical reasons. The VST normalization used previously is debated as I noted, but should be fine to use for the diversity analyses above. However when comparing relative abundances of groups, it's been shown that the compositional-nature of these data causes problems, that VST is not robust, and that reference frames (e.g., ALR, ILR, CLR transforms) or similar should be used (see https://www.nature.com/articles/s41467-019-10656-5).
- Figure 4 - I suggest to map instead the log-ratio of leaf to soil for each taxon, per my concerns re: comparing read counts among groups above.

Validity of the findings

Results:
- My only concern surrounds the analysis of VST-normalized read counts in quantitative analyses comparing groups (i.e., I have no concerns when they are used for typical diversity analyses or presence-absence/qualitative analyses). It has been shown that the compositional nature of next generation sequence data causes problems in interpreting 'changes in relative abundance' from sample to sample (see https://www.nature.com/articles/s41467-019-10656-5). Therefore I recommend to use reference frames, or log-ratios, if possible. I have provided specific suggestions where I was able to above.

Additional comments

Thank you for preparing such a wonderful article! It was a pleasure to read and the scope and findings are very interesting. I look forward to continuing to follow your work.

·

Basic reporting

Overall, the paper could use some stream lining to make the authors' message clearer to the reader. I have outlined some issues below, as well as in the "General comments to the author".

The authors seem to switch their vocabulary readily making it hard to follow. For example, in the introduction the authors use habitat breadth, habit specificity, and habitat occupancy, but do not define what they consider habitat to be. Is it solely soil versus phyllosphere? This seems more like substrate rather than habitat. Use of habitat here has issues as habitat is the environment of an organism and is an amalgamation of different factors (plant species, soil conditions, climate, co-occurring organisms). I think it would benefit the paper if you outlined major differences in the environment of the phyllosphere (UV, low nutrients…; Vorholt 2012 outlines this well) versus bulk soil (highly heterogeneous environment, high competition…). What do fungi need to be able to survive in one, the other, or both? Why would you expect some fungi to be "specialized" to soil or the phyllosphere?

In considering habitat breath and range size, the authors do not define what they mean by these terms. High throughput sequencing data limits one’s ability to detect habitat breadth and range size as it amplifies all target DNA in a sample which includes DNA from living and dead cells, as well as hyphae and spores. It is impossible to know what life stage a fungus is at and what substrate it is using based on sequencing data alone. It seems that here the habitat breadth and range size only include where the fungus has been detected. As a suggestion, you could use FunGuild to determine broad lifestyles of fungi providing possibly more insight into what they are doing.

Experimental design

Please be more explicit in your methodology for HTS as the quality of HTS data is dependent on the how sample libraries were made and for reproducibility of the results.
a. For example, you do not state whether library preparation was a one-step or two-step process and whether PCRs were done in triplicate to account for stochasticity in PCR amplification (see comment below).
b. Negative controls during extraction and PCR are not mentioned until they are referenced for contaminant removal (lines 154 to 156). Did you include positive controls in your sample preparation and workflow? Positive controls or mock communities allow for assessment of PCR and sequencing bias, additionally they allow assessment of the validity of OTU delimitation (see Bakker et al. 2018; Taylor et al. 2016; Nguyen et al. 2016). If mock communities were not used, then that should be explicitly stated as there is no way to verify that read abundance matches biological abundance, so analyses in which read abundance is used (even if normalized) are questionable. This is due to different copy numbers of rRNA genes in different fungal lineages (see Bakker 2018).
c. An additional issue is that no filtering of OTUs was done and singletons were kept, although this was not mentioned until the results section. It is generally recommended that OTUs with low read abundance and particularly singletons should be filtered out due to the change of errors when sequencing to that depth (see Edgar, 2013; Tedersoo et al. 2010; Dickie 2010; Nguyen et al. 2016). I would recommend verifying results without singletons or low abundance OTUs and including these results alongside the ones reported in the manuscript.
d. You don’t mention read normalization until Line 216 of the results. Please explicitly state how you normalized reads in your methods. Here I am assuming by normalizing the reads you mean rarefaction. Did you use size-based rarefaction or coverage-based rarefaction (Chao & Jost 2012)?

Please see more specific line-by-line comments in the "General comments to the author section" and broader comments in the "Validity of the findings".

Validity of the findings

I have concerns regarding assumptions that are made in the paper. For one, the authors state “Fungi are cosmopolitan…” (line 40) and “…so many are habitat generalists” (line 65-66). While yes fungi are found everywhere, specific species and groups of fungi show strong distribution patters that align with not only their symbiotic status (plant-associated, animal-associated, lichen-associated, free-living…), but also the abiotic and biotic environment (see Tedersoo et al. 2012; U’Ren et al. 2012; Bahram et al. 2013; Huang et al. 2018; U’Ren et al. 2019). As part of this, I think the models that are used lack important aspects of the environment that might add more depth to their results at both larger and smaller spatial scales. This includes host species (see below for specific question regarding this), geographic location (two distinct islands were sampled), forest type, and climatic variables (precipitation and temperature; this may correlate with geographical location if the main differences are between the islands). Lastly, limitations to the methodology are not clearly stated in the discussion section. Sampling was conducting by swabbing leaf surfaces and soil followed by metabarcoding. There is no way to distinguish what a particular OTU was doing (actively growing, dead, or a dormant spore) when it was sampled, so there is a huge caveat on the distribution patterns. Do these patterns represent the true distribution of fungi or the dispersal ability of fungi?

Please see more specific line-by-line comments in the "General comments to the author section".

Additional comments

In this paper, the authors calculate the range size of specialist and generalist fungi from both the bulk soil and phyllosphere using data from metabarcoding of the ITS1 region. In all, I found this paper interesting and novel in its goal. As the authors state, the calculation of range size of fungal species is an underexplored area, and the authors use a variety of methods to answer their main questions. The inclusion of two methods for analyzing distribution patterns (the area-occupancy analysis and semivariograms) was compelling, and the authors do a good job outline the differences between the two methods. I think this paper would be of interest to both mycologists and ecologists.

Line edits:

Line 63: change “specie’s” to species’

Line 65: change “withing” to within

Line 65-69: I disagree with the authors’ assessment that many fungi are habitat generalists, and that fungal species are cosmopolitan. It harkens back the Bass Becking hypothesis “Everything is everywhere, but the environment selects”. The authors provide no evidence to support these statements other than the occurrence of one species Aspergillis niger in the GlobalFungi database. This example has some inherent issues and I don’t think it can be considered a cosmopolitan species as it is a species complex and requires use of multiple loci to delimit species (da Silva et al. 2020). The GlobalFungi database uses records from HTS and includes only ITS sequences (Větrovský et al. 2020). HTS detects all fungal structures present in a sample, including spores. Aspergillus spp. are known to produce prolific numbers of aerial dispersed spores. It is impossible to know the life stage and function of a fungus isolated via HTS techniques.

Line 103-113: The final paragraph of the introduction does outline the goal of the study but is difficult to follow. It seems that the statistical analyses and results are outlined with subheadings that are meant to divide the analyses by the question they are meant to answer. Is that right? If so, the final paragraph is not written in a manner that it is easy to match the analysis and results to the question it is addressing.

Line 124: Are all the six transects in the exact same forest type, same soil type, same climate? They are across two different islands, but how similar are the habitats between islands?

Line 126-129: Do you have a reference for this sampling methodology that is swabbing leaf surfaces and soil? I have never heard of taking swabs of soil. The Qiagen DNeasy PowerSoil DNA isolation kit usually requires a sample of up to 0.20-0.25 g of soil. Where you able to isolate that much soil with a swab?

Line 131-132: Please change this sentence to “Plant host identity influences phyllosphere mycobiome..” and put proper citations. Also, here do you mean you swabbed multiple surfaces with the same swab? Or were multiple swabs taken, one for each host? If the latter, then were these extracted separately or together?

Line 142-143: Where PCRs conducted in triplicate? This is important to remove some of the natural stochasticity present in PCR amplification (see examples in Daru et al. 2019; Glassman et al. 2017; Bakker 2018). Was this a two-step PCR in which barcodes and Illumina adapters were put on in the second outer PCR?

Line 151: Did you try to make contiguous sequences out of the forward and reverse reads prior to discarding the reverse reads? Reverse reads are generally going to be lower quality than forward reads, but do provide value as they verify the lower quality end of the forward reads.

Line 151: How many chimeras were removed (total number and % of reads)?

Line 152: Why did you cluster at 97% sequence similarity?

Line 156: What percentage of OTUs could not be assigned to a fungal phylum?

Line 166: Change “Permutational” to permutational

Line 167: What were your explanatory variables in the PERMANOVA?

Line 171 to 173: Please move this sentence (“Fungal communities…to the soil below”) to the introduction.

Line 178: Why did you use a student T-test here? You conducted the test at two different spatial scales. Why not use an ANOVA? That would reduce you change of a type I error.

Line 188: What is S2?

Line 196-198: Please move this sentence (“Range size…transect.”) to the introduction.

Line 242-243: Please move sentence (“OTUs were first categorized…for each OTU.”) to the methods.

Line 249-250: Please move sentence (“Spatial autocorrelation…a Mantel test.”) to the methods.

Line 252-260: Please move section (“Range sizes were modeled using…were removed from downstream analysis.”) to the methods. Please not that I was unable to find Table S3. Do you mean Table S1?

Line 266-267: Please move sentence (“Including the culled…range-size comparisons.”) to the methods.

Line 272-273: Please move sentence (“To visually illustrate…but of the same transect.”) to the methods.

Line 289: Change “fugal” to fungal.

Line 325: Remove “1)” and “2)” (line 328). These are in separate sentences which makes the numbers confusing.

Line 327: Change “Fig.s” to Fig. Change “axis” to axes.

Line 369: “This result this study” should be changed.

Line 382: Here in your conclusion, you mention physical and temporal characteristics, but you have not outline how the phyllosphere and soil environments differ. Also, I am unsure of the temporal characteristic except in Fig. 1a in which you state that axis 2 corresponds to differences in the sampling times and islands, but sampling time was not including in any other analyses, correct? I do think it would improve the depth of your results if you included more physical and even the temporal aspect in your analyses as outlined above.

Figure 1a: In the legend, you mention that axis 2 corresponds to differences in sample date and island. Was this included in your PERMANOVA?

Figure 1b: Not referenced in your manuscript. Also, this is not particularly informative, and you may consider either modifying it or removing it.

Figure 3: Please change the titles of the graphs to match the habitat types you are using (soil and phyllosphere). Changing them here to ground and understory is confusing. If these represent other groups than soil and phyllosphere then it should be explicitly state. Also in this figure you mention that the outliers can be found in Supplemental Table S1, but in the manuscript you state that it is found in Table S3.

Figure 4: Very interesting figure. Could you place the names of the OTUs in the legend or in the manuscript, so they can be cross referenced in Table S1?

Figure 5: What is the y-axis? I understand that it is normalized OTU reads, but the axis range (0 to 10) doesn’t seem to match that.

Supplementary figure S1: Figure is cut off. Also, it would be helpful if the transects were labeled to match Table 1.

Supplementary Table S1: Why are all the OTUs called OTU97_***?? In Table S1, the outliers removed for Figure 3 should be clearly marked.

In the supplemental material, there is a flowchart, but I couldn’t find a reference to it in the manuscript. I think it is useful for understanding the workflow though.

Do you have plans to deposit the reads into the Sequence Read Archive?

---

## Round 0.2 · accepted · Accept

Thank you for your thoughtful and detailed responses to the reviewer's comments. I believe that you have sufficiently addressed their concerns and I am pleased to accept your manuscript for publication. However, as a note I wanted to mention that although you define habitat specialization differently, my previous work has addressed a similar question about links between fungal specialization and geographic distributions (U'Ren et al., 2019 Nature EE). In that study endophytes that were more specialized on particular host lineages had more limited geographic distributions. It is interesting to see your data also support this pattern despite differences in sampling approaches.

Reviewer 1 ·

Basic reporting

no comment

Experimental design

no comment

Validity of the findings

no comment

Additional comments

I thank the authors for carefully addressing my concerns. I don't have further comments and suggestions.

·

Basic reporting

Review for Milazzo et al. 2020 revision. High-throughput metabarcoding characterizes fungal endophyte diversity in the phyllosphere of a barley crop. Phytobiomes Journal.

In this paper, the authors calculate the range size of specialist and generalist fungi from both the bulk soil and phyllosphere using data from metabarcoding of the ITS1 region. In all, I found this paper interesting and novel in its goal. As the authors state, the calculation of range size of fungal species is an underexplored area, and the authors use a variety of methods to answer their main questions. The authors’ revisions greatly improved the paper and clarified their methods.

The authors have clarified some of the vocabulary they are using with the exception of niche and niche breadth. Niche most commonly means the environmental conditions and resources required for an organism to persist (as Elton and Hutchinson defined it), but can also mean just the environment. In this paper, it seems that niche means only the physical limits tolerated by species of fungi. Niche breadth is an important aspect of their conclusions, so how the authors are defining it here should be explained explicitly.

Line 270: Change “were” to was

Fig. 1: For 1a, you mention that separation is along axis 2. Are the points from each island well mixed or separated? Could you mention this or possibly change the color or shape of the points based on which island the sample was from?

Fig. 5: You do not have labels for panels (a and b) but refer to panel b in line 399.

Supplemental Figure S1: Figure is labeled S2 in file.

Experimental design

Methods are well clarified.

Validity of the findings

No comment.